# Empowering Patient Similarity Networks through Innovative Data-Quality-Aware Federated Profiling

**DOI:** 10.3390/s23146443

**Published:** 2023-07-16

**Authors:** Alramzana Nujum Navaz, Mohamed Adel Serhani, Hadeel T. El Kassabi, Ikbal Taleb

**Affiliations:** 1Department of Computer Science and Software Engineering, College of Information Technology, UAE University, Al Ain P.O. Box 15551, United Arab Emirates; 2College of Computing and Informatics, Sharjah University, Sharjah P.O. Box 27272, United Arab Emirates; mserhani@sharjah.ac.ae; 3Faculty of Applied Sciences & Technology, Humber College Institute of Technology & Advanced Learning, Toronto, ON M9W 5L7, Canada; hadeel.el-kassabi@humber.ca; 4College of Technological Innovation, Zayed University, Abu Dhabi P.O. Box 144534, United Arab Emirates; ikbal.taleb@zu.ac.ae

**Keywords:** federated learning, federated profiling, patient similarity network, federated patient similarity network, data quality profiling, deep learning, edge computing, eHealth

## Abstract

Continuous monitoring of patients involves collecting and analyzing sensory data from a multitude of sources. To overcome communication overhead, ensure data privacy and security, reduce data loss, and maintain efficient resource usage, the processing and analytics are moved close to where the data are located (e.g., the edge). However, data quality (DQ) can be degraded because of imprecise or malfunctioning sensors, dynamic changes in the environment, transmission failures, or delays. Therefore, it is crucial to keep an eye on data quality and spot problems as quickly as possible, so that they do not mislead clinical judgments and lead to the wrong course of action. In this article, a novel approach called federated data quality profiling (FDQP) is proposed to assess the quality of the data at the edge. FDQP is inspired by federated learning (FL) and serves as a condensed document or a guide for node data quality assurance. The FDQP formal model is developed to capture the quality dimensions specified in the data quality profile (DQP). The proposed approach uses federated feature selection to improve classifier precision and rank features based on criteria such as feature value, outlier percentage, and missing data percentage. Extensive experimentation using a fetal dataset split into different edge nodes and a set of scenarios were carefully chosen to evaluate the proposed FDQP model. The results of the experiments demonstrated that the proposed FDQP approach positively improved the DQ, and thus, impacted the accuracy of the federated patient similarity network (FPSN)-based machine learning models. The proposed data-quality-aware federated PSN architecture leveraging FDQP model with data collected from edge nodes can effectively improve the data quality and accuracy of the federated patient similarity network (FPSN)-based machine learning models. Our profiling algorithm used lightweight profile exchange instead of full data processing at the edge, which resulted in optimal data quality achievement, thus improving efficiency. Overall, FDQP is an effective method for assessing data quality in the edge computing environment, and we believe that the proposed approach can be applied to other scenarios beyond patient monitoring.

## 1. Introduction

As the internet of things (IoT) became more pervasive, it is evident that big data capabilities are undergoing a revolution, with enhanced domain sensing capabilities. Nevertheless, many IoT-related projects are hampered by real-time connectivity issues and insufficient computing power to handle the ever-increasing volume of information processing. Furthermore, limitations in data transport capabilities further exacerbate these challenges, necessitating the execution of complex data analysis on heterogeneous computing platforms. These constraints impose a considerable risk of information loss during data processing, particularly when employing aggregations, approximations, and filtrations to overcome resource limitations. This, in turn, has a direct impact on data accuracy, as the outcomes of data processing become susceptible to inaccuracies and uncertainties [1]. In addition to environmental factors, challenges during data creation and collection at the data sources add to the complexities, as IoT generates a massive volume of data that needs to be efficiently collected, stored, processed, and analyzed. Factors such as reduced sensor precision, communication latencies, short battery life, and limited availability of sensor/actuator sets contribute to diminished data accuracy. Additionally, the potential for data breaches and privacy concerns due to the vast amount of sensitive data collected and transmitted further compounds the need for robust security measures to protect the integrity and confidentiality of the collected data. These challenges must be addressed to ensure the successful implementation and operation of IoT projects, particularly in domains where data accuracy, real-time analysis, and efficient data transport are crucial. To optimize IoT and mobile edge computing (MEC) architectures, techniques such as feature selection and data fusion can be leveraged, while also considering strategies for energy conservation and energy harvesting [2]. These approaches aim to improve the quality of low-cost sensor data, enhancing the overall performance and reliability of IoT systems [3]. By overcoming these obstacles, IoT projects can unlock their full potential and effectively contribute to data-driven decision making in various domains.

In the field of healthcare, the influence of data quality (DQ) flaws on physician judgments decreases the likelihood of patients receiving optimal treatment, jeopardizing their health and well-being. While poorly designed DQ attributes result in miscalculations and misinterpretations that can have significant negative effects on healthcare providers and patients, only a few studies document the actual effects of DQ flaws on patient care decisions. This includes reduced validity of critical clinical characteristics and incomplete data, which affects the propensity to prescribe medications or perform invasive procedures [4]. Consequently, it is essential to measure the impact of DQ issues on clinical decisions and emphasize their relative importance. DQ issues must be carefully addressed at the source and effectively managed before they can affect the clinical decision-making process in order to preserve the integrity of the generated data and support the highest standards of clinical decision making.

Gartner [5] estimated that 60% of organizations will utilize machine-learning-enabled DQ technology by 2022 to reduce human operations for data quality improvement, and 50% will use data quality solutions by 2024 to promote digital business. DQ is critical for tracking data value and relevance, and we believe its use in quantifying data will give us a handle on what data are available, what the data’s value might be for business decision making, and whether the data should be assessed primarily during the data transformations at the pre-processing and processing stages of the data. The accuracy of a classification model is heavily dependent on the DQ, so measuring DQ [6] is critical for estimating task complexity earlier. DQ attributes should be verified, improved, and regulated throughout their life cycle, as they have a direct impact on the conclusions drawn from data analysis. To capture the quality requirements, characteristics, dimensions, scores, and applications of quality rules, data profiling [7,8] has become a popular approach. DQ assurance and this approach have become so intertwined that they are often referred to as the same. It is a collection of techniques used to facilitate a variety of data management tasks, such as data quality evaluation and metadata management. In healthcare data, on the other hand, ensuring DQ is a time- and resource-intensive process, especially when dealing with large amounts of data. The FL method may be crucial, as it allows for the construction of a common quality model based on multiple sources without sharing data. This method facilitates the need to combine multiple data sources to ensure the quality of data analytics while also protecting the privacy of each individual and reducing the transfer time for data. Our proposal to use a federated data quality profiling model to ensure the privacy and security of eHealth data, and is motivated by the need to address DQ concerns at every stage of the data’s lifecycle, primarily at the edge.

### 1.1. Background

Any work on DQ is incomplete unless the DQ measures and metrics are stated, as they are critical components in measuring data quality. To familiarize the reader with the concept, some PSN background information is also provided.

#### 1.1.1. Data Quality Dimensions and Metrics

For a given situation, some data may be more important than others in terms of achieving the strategic vision, and therefore, when it comes to data quality (DQ), it is important to focus on the most significant data. An entirely new set of metrics incorporating “data weights” has been proposed by the authors in [9]. Choosing a set of dimensions to work with is an important part of the approach, and to measure each dimension, a metric must be selected. To accomplish continuous improvement, TDQM (total data quality methodology) [10] is one of the few approaches that operate in a cyclical or revolutionary fashion. When it comes to metrics, the only thing it uses is “basic percentages”, such as the percentage of missing data for the “completeness” dimension. Data quality assessment (DQA) combines subjective and objective evaluations. The root causes of discrepancies can give great insight into the DQ problems and guide data quality improvement efforts. According to the authors of [11], the five requirements of data quality metrics are the existence of minimum and maximum metric values (R1), interval scaling of the metric values (R2), and quality of the configuration parameters, as well as the determination of metric values (R3), sound aggregated metric values (R4), and economic efficiency of the metric values (R5). The term dimension is used to describe elements of data that can be measured, as well as the ways in which data quality can be evaluated and quantified. Data accuracy, completeness, uniqueness, timeliness, and validity are the six primary criteria that determine the quality of data [12]. The following are the accepted definitions of each of these metrics.

Accuracy: The degree to which data correctly portray the thing or event under investigation in the “actual world”. Validity is a quality dimension linked to accuracy.

Completeness: The ratio of data saved to the potential for “completeness”. Validity and accuracy are two quality dimensions related to completeness.

Uniqueness: No object, regardless of how it is identified, will be recorded more than once. Consistency is the quality dimension connected to uniqueness.

Timeliness: The degree to which data correctly represent reality at a given point in time. The quality component associated with timeliness is accuracy.

Validity: Data are accepted if they adhere to the syntax of the specification (format, type, and range). The validity-related quality dimensions include accuracy, completeness, consistency, and uniqueness.

It is also important to note that the related literature revealed that most DQ metrics are strongly associated with one another. Data quality characteristics appear in the data collection process as well as the preprocessing stage—this includes both the upstream and downstream stages of the data processing [13]. The upstream influencing factors are determined by the data collection system. The loss of data quality is expressed by missing values in the event of data storage failures or the inability to measure the requested physical values. The completeness indicator considers any missing values. Accessibility, mobility, and recovery all fall under this umbrella term. The data analyst cannot use the signal data if it cannot be accessed or if it cannot be transferred to a database or data mining software. In the event of a failure or loss of data, a lack of recoverability results in a lack of information. When it comes to traceability, the impact is not caused by missing values in the time series, but rather by a lack of details about the dataset itself. The various subdimensions of completeness also cover this influence. When compared to the factors that have an impact on data quality upstream, the factors that have an impact on downstream DQ that appear during data preprocessing are accuracy, credibility, consistency, and relevance [13]. It is also worth noting that the conversion of signal data to the international system of units (SI), for example, does not have a negative impact on the quality of the data. Compliance is a problem for data quality if there is a lack of information that prevents conversion to a particular standard. Different definitions for data quality metrics have been presented by many research studies. A few of the metrics are listed in the following sections.

#### 1.1.2. Timeliness Metric

The ratio between currency and volatility determines the timeliness and they must be measured using the same units of time. Time tags provide information about the date the data item was acquired. For example, highly volatile data, such as stock quotes or currency conversion tables, have a very short shelf life. Depending on when the information product is delivered to the customer, an information product’s timeliness can vary. The data quality metric for timeliness is defined by Ballou et al. in [14] as follows: $Timeliness={\left[(1-currency/volatility),0\right]}^{s}$. The parameter age of the data value represents the elapsed time between the real-world event’s occurrence (i.e., the time the data value was created in the real world) and the determination of the data value’s timeliness. The maximum amount of time that the values of the considered attribute will remain current is defined as the parameter shelf life or volatility. In other words, a higher value of the metric for timeliness implies a higher value of the parameter volatility and vice versa. The metric’s sensitivity to the ratio age of the data value depends on the exponent s>0, which must be determined based on expert estimations.

#### 1.1.3. Completeness Metric

Complete data have been described as data with all values recorded. Missing data can typically be indicated by null or another indicator in most applications. The metric for completeness is defined in [15] as: Completeness=1−(MT/NK), where MT is the proportion of tuples in relation with null values to the total number of tuples and NK is the total number of tuples.

#### 1.1.4. Correctness Metric

Heinrichs [11] defined correctness as a metric to evaluate the accuracy of a stored data value: $Correctness=1/(d\left(w,w_{m}\right)+1)$, where *w* is the stored data value, wm is the corresponding real-world value, and *d* is a domain-specific distance measure.

#### 1.1.5. Patient Similarity Network (PSN)

PSNs are designed to assess whether a patient is likely to benefit from treatment modalities and lifestyle changes of other patients who are likely to be similar to the current patient. The objective of PSNs is to recommend the appropriate therapy and medicine for the patient based on aggregated data extracted from other patients with similar characteristics. A few of the PSN challenges are listed as follows. Clinical narrative data that are diverse and heterogeneous enrich hidden information that is useful in selecting the most comparable patients. Medical occurrences are time-sensitive, and understanding the dynamics of medical terminology and conclusions requires temporal information. When using noisy clinical datasets, interpreting temporal representation is highly challenging, and the result prediction accuracy is low. The dimensionality of health datasets is varied and high. For example, the electronic health record (EHR) stores a wide range of data, such as diagnoses, drugs, laboratory tests, and X-rays, as well as medical events like diseases and treatments. Because the data are a mix of static and dynamic, modeling and processing are difficult.

### 1.2. Motivation

The main motivation of this paper is to address a few of the challenges with respect to data quality. The heterogeneity of eHealth data from diverse data sources may be addressed using the generalized hybrid PSN model proposed in our paper [16]. The model is effective in solving big data challenges when patient cases contain both structured and unstructured data by employing an autoencoder to enforce dimensionality reduction in the model. The patient similarity network fusion strategy uses PSN distance estimations from static and dynamic data to emphasize patient pair similarity and reduce interference produced by non-similar pairings. However, this PSN fusion strategy was designed with processing at the centralized server, and not considering the quality of data received from the edge nodes or the source.

Through experiments, we assessed the influence of individual edge data quality on FL model accuracy, which motivates us to investigate data quality-aware edge selection and profiling for PSN, integrating it with FL services to address faulty data issues at dispersed client sources. FL training may use a lot of computer resources when there are a lot of training datasets and jobs, and therefore, we propose a model to efficiently improve data quality in remote learning clients. We create a profile based on the data context that can be dynamically sent to all FL clients, and we execute client data selection and augmentation to significantly reduce patient data transmission. When compared to cutting-edge data quality enhancement approaches, our proposed model can significantly improve FL performance for a wide range of learning tasks and FL scenarios. In this paper, we propose the DQA FDQ profiling model, a quality-driven, edge-based federated strategy for sensor-based monitoring setup that is motivated by the following:Capture quality at the beginning of the data acquisition process to ensure that DQ is maintained throughout the data lifecycle.In the event that quality assessment criteria are dynamically updated in the case of real-time data, it is recommended to introduce a data quality profile, abbreviated as DQP, to support quality assessment at each edge node.The federated DQP can provide a more robust and detailed quality evaluation because it will be able to capture the vast majority of quality issues occurring across all nodes. Adopting a strategy to eliminate edges with noisy data and facilitate client selection will reduce the impact of low-quality data on model training. Thus, the federated quality profile will exclude edges with greater quality profile variation.Federated profiling will have a low overhead because it will focus only on the quality profile measures and variance and not on the entire datasets stored at the various nodes.

This article aims to address the challenges of data quality in healthcare by proposing a novel federated data quality profiling (FDQP) approach. The proposed FDQP model in federated PSN evaluates the quality of patient data obtained from edge nodes and enhances the accuracy of machine learning models through profiling algorithm and federated feature selection. Experimental results demonstrate the effectiveness of federated profiling in improving data quality and accuracy. The contributions of this paper are described in the following section.

#### Contributions

Pioneering the federated data quality profiling (FDQP) technique for evaluating patient data quality: This paper introduces a novel approach, referred to as federated data quality profiling (FDQP), which aims to evaluate the quality of patient data obtained from edge nodes. By pioneering this technique, the paper addresses the need for assessing the reliability and accuracy of decentralized healthcare data.Development of an FDQP formal model to capture quality dimensions: To encapsulate the various dimensions of data quality profile (DQP), the paper establishes an FDQP formal model. This model serves as a comprehensive framework for representing and analyzing the different aspects of data quality, contributing to the overall understanding and evaluation of the patient data’s quality characteristics.Utilization of federated feature selection for enhanced precision: By leveraging federated feature selection techniques, this research enhances the precision of classifiers used in analyzing patient data. The paper categorizes features based on metrics such as feature value, percentage of outliers, and missing data percentage, thereby improving the accuracy and reliability of the classification process.Extensive experimental evaluation of the FDQP model: The proposed FDQP model undergoes an extensive series of experiments, utilizing a distributed fetal dataset across diverse edge nodes and varying scenarios. This rigorous evaluation enables the assessment of the effectiveness and performance of the FDQP model in practical healthcare settings.Improved data quality and accuracy in FPSN-based machine learning models: The study demonstrates a noticeable enhancement in data quality and accuracy of federated patient similarity network (FPSN)-based machine learning models as a result of adopting the FDQP approach. By incorporating the FDQP model, the paper shows how the proposed methodology positively impacts the performance of FPSN-based models in healthcare data analysis tasks.Proposal of a data-quality-conscious federated PSN architecture: This paper presents a novel architecture, termed the data-quality cognizant federated PSN architecture, which integrates the FDQP model to effectively improve the data quality and accuracy of FPSN-based machine learning models. This proposed architecture addresses the challenge of utilizing data collected from edge nodes, while ensuring high-quality and reliable healthcare analytics.Application of an efficient profiling algorithm for data quality optimization: The research applies a profiling algorithm that prioritizes efficient lightweight profile exchange over complete data processing at the edge. By adopting this approach, the paper advocates for optimized achievement in data quality, allowing for streamlined data processing and improved efficiency in healthcare data analysis workflows.

The FDQP approach has the potential to be applied in a range of scenarios beyond patient monitoring. Here are a few scenarios that highlight the broader applicability of FDQP:

Industrial automation: In industrial settings, where large amounts of sensor data are collected from various machines and equipment, FDQP can be used to assess the quality of data to ensure accurate decision making and optimize production processes.

Environmental monitoring: FDQP can play a crucial role in evaluating the quality of environmental sensor data, such as air quality measurements, water quality parameters, and climate data. This can aid in monitoring and addressing environmental issues effectively.

Smart cities: With the increasing adoption of IoT technologies in smart city applications, FDQP can be employed to evaluate the quality of data collected from various sensors deployed throughout the city. This can support better urban planning, resource management, and citizen services.

The versatility of the FDQP approach can be extended to diverse IoT applications, enabling data accuracy and integrity across various domains for reliable decision making, improved operational efficiency, and enhanced outcomes.

The paper is organized as follows. In Section 2, we review prior research on DQ and FL. Section 3 describes our FDQ profiling model and proposed algorithm, while Section 4 describes the evaluation of our model using the fetal health monitoring dataset and includes the evaluation methodology and experiments to illustrate the benefits of our approach. Section 5 discusses the experiment findings, underlying principles, and limitations, to present a balanced and realistic view of the proposed approach. Finally, Section 6 concludes the paper and points to some future research directions.

## 2. Related Work

Many challenges must be resolved when managing a federated set of health data, including individual user rights and needs, maintaining user data privacy, establishing rules for creating and changing accounts and roles, providing tools for sharing private data with selected audiences, and allocating shared costs among participants in a collaborative effort, including data sharing, transfer, and integration. Clinical information can be protected by using FL [17], as data transfer is not required and all necessary information can be stored locally in a patient’s healthcare organization or the patient data edges. In the data analytics lifecycle, however, erroneous data discovered at one stage may have originated from the processing performed upstream. It is essential to ensure the integrity of each link in the lifecycle in order to determine the severity of a problem, as the ability to track the propagation of defects through a system can aid in determining its magnitude. Eliminating false positives, or results that do not pertain to a particular problem is another benefit of DQ profiling [7]. Consequently, managing DQ profiling through federation enhances DQ evaluation while maintaining data confidentiality. The following is a discussion of the current state of quality management over distributed and federated sensory data.

Some data quality objectives can be achieved through database integration using DQ attributes, such as querying the source database for inconsistencies and reconciling inconsistencies during merging using federated database approaches [18]. In global applications, the metadata models were used to identify DQ-related query targets and to dynamically combine data from component databases. The number of complex computations that can be accomplished on mobile devices has recently risen, supporting the spread of the “on-demand-edge-computing” paradigm, which is characterized by the concept of FL frameworks. The authors of [19] propose a DQ-centric big data architecture for federated sensor service clouds, in which high-quality data from a large number of separately managed sensors is exchanged or even traded in real time. One of FL’s challenges is training clients with poor or severely biased data, which may jeopardize FL’s efficiency and efficacy, resulting in a suboptimal global model that requires more rounds of computation to converge. Based on the FL technique, a data representation profiling method called FEDPROF [20] was proposed, which adaptively modifies clients’ participation likelihood based on their profile dissimilarity while maintaining data locality.

By utilizing FL based on source data instead of centralized data, mobile edge computing becomes more intelligent. The additional computation may be used in FL to minimize the number of communication rounds necessary to train the model. For example, the in-edge AI framework [21] relies on edge caching and computation offloading to boost computation and communication per edge device as an effective technique to handle the various challenges related to energy consumption, privacy, and modeling fairness at the edge. All stakeholders in the underlying data processing edges can have fair access to high-quality data when using FL methods. When an attack is detected, FairFed [22], a new FL framework, rejects model updates that lead to an attack.

Although there are studies that highlight FL, DQ, and PSN separately, none of these initiatives explored DQ-aware federated PSN to assess the quality of the data itself and its impact on increasing the prediction accuracy of PSN models. In the following section, we highlight the related concepts within the context of FL based on the literature review.

### 2.1. Data Quality Assessment

Many approaches for detecting noisy data have been proposed in the literature, with the most common three being distance-based, density-based, and clustering-based strategies [18,23]. In its most basic form, an outlier is a value that differs from the rest of the data in a collection. Outlier detection assigns each object an outlier score that indicates how much it deviates from the norm. Such algorithms can remove any specified amount of noise, for example, by sorting objects based on their “outlier score” and deleting the items with the highest outlier scores until the appropriate percentage of objects is removed. Data profiling [24] can solve DQ concerns, such as noise and outliers, inconsistent data, duplicate data, and missing values [23,25,26].

Due to the presence of minor class representation in the data, class imbalance problems arise, and when the data are classified, the classifiers are biased toward the dominant class. For instance, in medical diagnosis, the detection of cancerous cells and the misclassification of non-cancerous cells may necessitate additional clinical testing. However, if the misclassification of cancerous cells leads to the prescription of no additional tests, the patient’s cancer may go undetected, which poses a significant health risk. In the medical diagnosis example, however, the classifier might very well mislabel some benign tumors as malignant (cancerous), thereby subjecting a patient without cancer to unnecessary tests. Similarly, there are many instances where the overall classification accuracy of ML algorithms misclassifies minority classes. This leads to increased misclassification costs, time, and risk assessment. Each client has a unique quantity of locally stored training data. Therefore, the reliability of the trained values will vary depending on the client, as there is a possibility that the training sessions will be too brief [27].

In today’s information age, where vast amounts of data are being generated at rapid intervals, it can be difficult to sift through this flood of information to find the bits of value. There is a centralization issue, because large tech companies control the majority of the data and resources required to effectively train machine learning models. Traditional machine learning settings explore the subject of relevant data selection under the presumption that all relevant data are available in one location for computation. Due to the fact that the data in FL settings are dispersed among numerous clients and the server is unable to inspect it because of privacy restrictions [28], the conventional methods for choosing pertinent data are not applicable in this situation.

An approach that finds relevant data and stores it in a decentralized way with an innovative approach to network sharding that the authors call the interest group [29]. The data added to the ledger are verified for accuracy by the proof of common interest. In [28], it is suggested to use a method called federated learning with relevant data (FLRD), which allows clients to select data relevant to the server’s objective, leading to improved performance of the global learned model. Each client must learn a function that estimates data point relevance without compromising privacy.

Some of the challenges in federated data quality profiling are inspired by the FL environment, which involves data that are distributed and from several sources, which presents its own set of issues, as opposed to a centralized dataset that is simply dispersed around worker nodes.

### 2.2. Quality-Aware Client Selection in FL

A strategy for user recruitment called quality-aware user recruitment [30] has been addressed to handle the problem of optimizing the observed DQ in mobile crowd sourcing (MCS) mode. Through federated learning, this work predicts the quality of sensed data from various users by examining the correlation between data and context information.

The majority of current FL approaches focus on the supervised setting, where each client’s data are labeled. FedTriNet [31] is proposed as a federated semi-supervised learning method with a pseudo-labeling method having two learning phases, because it is impossible to completely label client data in real-world applications.

One of the most prized qualities of a data valuation metric is its fairness. Data owners are key to federated learning’s success. Each data owner’s contribution to the model’s performance should be reflected in the FL setting’s data valuation metric. Furthermore, it is important to fairly evaluate data owners’ quality and contribution to the final model and reward accordingly if you want them to continue contributing. The federated shapley value [32] satisfies many data valuation properties. Two data owners with the same local data may not receive the same evaluation with a federated Shapley value. Completed federated Shapley value [33], based on a low-rank matrix completion formulation, improves the federated Shapley value’s fairness. Data source selection is required in practical applications due to the high cost and low availability of communication channels, and FDSS [34], an algorithm, is devised, which can effectively address this issue in both static and dynamic contexts and demonstrate that it can be transformed into a monotone submodular maximization problem.

An adaptive accuracy threshold aggregation strategy based on federated learning [35] is proposed, which can satisfy the practical needs of multi-party data learning without requiring the sharing of sensitive data. The local node obtains information from the local dataset. After local training, the aggregation node verifies that the conditions for this round’s aggregation have been met. In this instance, local model data are retrieved from the local node, and model aggregation is carried out. The algorithm calculates the optimal threshold to reduce the number of communications between local nodes and aggregation nodes. This algorithm can alter the accuracy of each round during model training and determine the optimal threshold. Since the local model data indirectly reflects the node sample information, an attacker can deduce the sample data from the effective model information, reducing the number of communications and the probability of privacy leakage. Hybrid logical security framework (HLSF) [36] proposes strong authentication and data confidentiality mechanisms, which improves security and network capabilities, promotes green IoT, and addresses privacy concerns.

AUCTION [37] is intended to embed the client selection policy into a neural network and then use reinforcement learning to automatically learn client selection rules based on observed client status and feedback incentives quantified by federated learning performance. The policy network is based on an encoder–decoder deep neural network with an attention mechanism that can adjust to dynamic changes in the number of prospective clients. TiFL [38], a tier-based FL system provides an adaptive client selection approach that uses quality as an indirect indicator to infer data heterogeneity information and updates the tiering algorithm on the fly. This minimizes training time while increasing accuracy. They also employed a lightweight profiler to evaluate each client’s training time and put them into logical data pools known as tiers depending on measured latency.

### 2.3. Data Heterogeneity on FL

Since data in the FL paradigm cannot be gathered or shared, data heterogeneity inevitably arises, severely impacting the efficacy of federated analytics. The analytics-driven client selection (FedACS) [39] framework proposed a three-pronged approach to dealing with the issue of data heterogeneity. Without revealing any private information, clients first generate insights about their local data; then, the server uses these insights to infer the situation of clients’ data using Hoeffding’s inequality. Finally, a client pool is formed by selecting individuals from a slightly more diverse set of data. Each individual customer has insufficient data because of costly labeling and the inability to store additional information. Large amounts of unlabeled data are readily available in the public cloud (e.g., social media). To boost DL model efficacy, Ada-FedSemi [40] uses both local and cloud-based labeled and unlabeled data. Due to the high volume and low availability of the clients, a carefully chosen subset of clients is used in FL during each iteration of training. The parameter server compiles these local models into aggregated pseudo-labels for unlabeled data, which is then used to train a better global model. Federated learning has been widely utilized to connect resource-constrained devices for neural network on-the-edge training. When federated learning deploys identical neural network models to heterogeneous edge devices, those with less processing capacity may dramatically delay the synchronized parameter aggregation, resulting in serious computational straggler problems [41]. These computational stragglers are edge devices with little processing capability that are attracting increasing attention from the research community [42]. Although training model improvement can help with stragglers, the optimized models frequently result in divergent structures due to diverse device resource limits, which has a substantial impact on collaborative convergence. FedProx [43], a variant of FedAvg [44], is presented to address system heterogeneity caused by system feature variability on federated training devices. System heterogeneity has a significant influence on model aggregation efficiency and accuracy, causing the optimization to diverge [45].

### 2.4. Implications of Communication in FL

Communication is an important part of FL and approaches to decrease communication, such as local updating and model compression, must be understood through a thorough analysis of the trade-off between accuracy and communication for each strategy. Bulk synchronous and asynchronous techniques are the two most often researched communication strategies in distributed optimization. These strategies are more practical in data center settings because worker nodes are generally dedicated to the task, i.e., they are ready to `pull’ their next job from the central node as soon as the results of their previous job are `pushed’. In federated networks, on the other hand, each device is typically unfocused on the task at hand, and most devices are inactive at any given time. As a result, it is worthwhile to investigate the impacts of this more realistic device-centric communication architecture, in which each device may select when to ’wake up’ and connect with the central server in an event-triggered manner [46]. To adapt the FL model to 6G networks [47] with huge, heterogeneous devices and networks, a communication-efficient approach must be developed, considering two major aspects: (a) lowering the total number of communication rounds or (b) reducing the number of gradients in each communication round. Another challenge is that the majority of present FL works use machine learning models with full-precision weights, and virtually all these models contain a high number of redundant parameters that do not need to be transferred to the server, spending an unnecessary amount of communication expenses. In addition, using data from edge devices is not always possible due to the devices’ inability to transmit data quickly enough or due to poor connectivity circumstances [48]. A ternary federated averaging protocol (T-FedAvg) [49] was developed to overcome this issue by reducing the upstream and downstream transmission of FL systems.

### 2.5. Data Model Quality-Aware FL

Another focus of recent research is to raise FL model quality awareness. The model updates contributed by computing nodes training with their local data determine the quality of federated learning. The model update qualities of computing nodes can vary dramatically depending on various factors (e.g., training data size, mislabeled data samples, skewed data distributions). The accumulation of low-quality model updates can degrade the overall model quality. A framework called FAIR [50] integrates three major components to achieve efficient FL learning: learning quality estimation, a reverse auction problem to encourage the participation of high-quality and inexpensive computing nodes, and auto-weighted model aggregation.

Federated edge learning’s privacy-preserving aspect makes it the future of wireless edge network training. In [51], data-quality aspects are used to schedule edges for collaborative training. First, the learning algorithm’s components, dataset diversity, and edge node reputation are defined. The authors proposed a DQ-based scheduling (DQS) algorithm that prioritizes reliable devices with rich and diverse datasets.

### 2.6. FL Data Streaming Concerns

There are several practical difficulties that occur when FL is used with real-life streamed data. Concept drift (underlying data-generation model changes over time), diurnal variations (devices exhibit different behavior at different times of the day or week), and cold start problems (new devices enter the network) are just a few of the issues [46] that must be handled with caution. The Adaptive-FedAVG FL [52] approach works with nonstationary data-generating systems that are impacted by concept drifts by modifying the learning rate to increase the learning phase’s flexibility, allowing it to adapt to concept drift.

Patient privacy and the risks associated with releasing personal health information, such as stigma, continue to be key challenges in healthcare. Because patient data are not exchanged between multiple locations, but the model goes to the data, our recommended DQA-FPSN model is created with privacy in mind. This enables the identification of similar individuals without risking patient privacy, allowing for accurate diagnosis and lifestyle recommendations.

Table 1 provides an overview of notable related work in the paper, highlighting different contexts and challenges addressed in the literature. It presents key features, advantages, and disadvantages of each reference, offering a concise understanding of the discussed work.

### 2.7. Research Gaps

Table 2 compares the proposed model, DQA FPSN, with state-of-the-art approaches in the field of FL. The listed approaches encompass various aspects, including node/client selection, dimension reduction, data heterogeneity, federated profiling, data quality assessment, FL-based methods, and accuracy Improvement. It is worth noting that the proposed DQA FPSN addresses several research gaps that have not been explored in the existing literature.

Despite the advancements in FL approaches, there is a lack of emphasis on quality-aware client selection, which plays a crucial role in ensuring the reliability and representativeness of the participating clients in the FL process. Another research gap lies in the dimension reduction or feature selection techniques specifically tailored for FL settings. Existing methods often assume homogeneous data, and there is a need to explore dimension reduction approaches that can handle the inherent data heterogeneity across different clients while preserving privacy and data utility. Current approaches focus on aggregating model updates, but there is room for improvement in terms of profiling the participating clients to better understand their characteristics and performance, leading to more informed aggregation strategies, while FL has shown promising results in various applications, ensuring accuracy improvement by the integration of DQ assessment, and DQ dimensions into FL frameworks is another area that requires more attention. There is a need for advanced optimization techniques and personalized learning approaches that can adapt to each client’s data distribution and enable more accurate and reliable model updates for heterogeneous data, specific to various domains, such as PSN, in healthcare. Addressing these gaps would significantly enhance the overall effectiveness of FL in real-world scenarios.

## 3. Data-Quality-Aware FPSN Model

In this section, we present our proposed data-quality-aware federated PSN (DQA FPSN) model, how to apply FDQP with an illustration, the detailed algorithm, and finally the model formulation. Figure 1 details our proposed DQA PSN model architecture that features federated quality profiling, where the data sources are at the edge. The architecture also features a cloud-based server that facilitates the profiling federation and the federated PSN score aggregation. Pre- and post-data quality evaluation is performed, and the process is repeated until the data quality reaches acceptable tolerance levels. The following are the sequential steps of the model processes.

1.In the initial stage, the centralized cloud server sends the baseline DQP to the edge nodes.2.Subsequently, at the edge local node, each node verifies and evaluates the data quality acquired from the sensors, updates the DQP, and transmits it back to the server.3.Finally, the server integrates the DQPs received from the edge nodes to create the federated data quality profile (FDQP), which is transferred to the edge nodes.4.FDQP is then applied to the local edge data that creates the quality-enriched data, which will be the basis for the PSN data fusion model at the edge.5.The resulting patient similarity score is sent back to the cloud server.6.The FPSN score aggregation model receives the model updates from the edges and the aggregation of the similarity scores takes place at the cloud server.7.The final patient similarity score is used as a basis to detect the most similar patient.8.DQ evaluation measures of performance such as accuracy are calculated.9.The process is further assessed with pre- and post-DQ evaluations.

### 3.1. Data-Quality-Aware FPSN Model Overview

An overview of the proposed model with FPSN and FDQP edge enhancement is shown in Figure 1. The objective of this framework is to enhance the precision and speed of data processing at the edge of the network.

### 3.2. Federated Data Quality Profiling

Figure 2 depicts federated data quality (FDQ) profiling using an example. The baseline quality profile specifies the needed data quality characteristics such as completeness, accuracy, and timeliness, as well as the data quality standards. Based on the baseline quality profile, we identify the dimensions of DQ with severe issues. A missing data-related rule, for example, will infer some actions (e.g., replace missing values with the mean) depending on the kind of data and the degree of tolerance specified in the DQP. This baseline DQP will be forwarded to the source edges, where the local dataset will be reviewed using the profile and a new quality profile will be constructed using some or all of the criteria. If the new profile satisfies the baseline quality profile, it is sent back to the server. It is worth noting that we are considering the edges of collecting and holding identical datasets with similar characteristics. Hence, Figure 2 shows that source edge 1 with attribute ID 1 has a missing value of 70%, and thus, rule 2.2 is used to eliminate the whole column.

Similarly, at each edge node, a local profile is relayed to the server node, which aggregates all profiles. According to the context and the rules, the aggregate process will use min, max, total, or average aggregation. The aggregated quality profiles will then be integrated to generate the federated data quality profile, which will include optimal rules based on attribute correlations. Furthermore, the feature selection rules will be defined in the federated data quality profile based on attribute priority and ranking. Thus, the federated data quality profile will have a well-defined purpose, quantifiable metrics, and optimized rules for selection, as well as explicit formulas for combining local/federated data profiles. Finally, the rules in the federated data quality profile are propagated and enforced across all nodes in the federation, ensuring enhanced data quality.

### 3.3. Model Formulation

Let *A* represent a collection of data attributes of the dataset expressed by $A=a_{0},\cdots ,a_{j},\cdots ,a_{D}$, where *D* is the number of attributes (or the dimensionality of the dataset) and aj is an attribute represented by its type, possible values, weight, tolerance, and rules. For each aj in *A*, the weights of each attribute are represented by a set $W=w_{0},\cdots ,w_{i},\cdots ,w_{p}$, where the sum of all weights should be 1. The weights are defined using a kernel function that provides the priority and importance of each attribute in the set *A*, thus enabling feature selection. Each attribute evaluation is mapped to a set of data quality dimensions $D=d_{0},\cdots ,d_{k},\cdots ,d_{q}$. *T* is a collection of minimum acceptable tolerance levels, $T=t_{0},\cdots ,t_{l},\cdots ,t_{r}$, established by data specialists and associated with every quality dimension (e.g., completeness accepted tolerance is 70 percent, which means the attributes need to comply with above 70 percent completeness). There are zero, one, or more applicable rules, $R=r_{0},\cdots ,r_{m},\cdots ,r_{s}$, to the attributes, which are applied when meeting criteria based on acceptable tolerance levels specified for the attributes. Rules can be tailored to handle quality issues and include corrective actions, such as data imputation or outlier elimination.

Our algorithm’s main phases are baseline representation profiling, edge representation profiling, and federated profiling. A DQP is, thus, represented by (A,W,D,T,R). First, the baseline DQP, DQPv is generated based on the sample dataset available at the server, which is similar to and characterized by independent and identically distributed (IID) datasets available at the edges. Quality requirements or preferences can be set for all attributes (the entire dataset) by default, as well as applied to individual attributes, on demand. The settings are determined by the data quality dimensions selected for profiling and by the requirements of the application. Certain dimensions, such as completeness, can be set for all attributes by specifying the expected ratio or tolerance that must be met. Several other dimensions, such as timeliness, are more focused on specific attributes such as time or date.

The DQPv is subsequently sent to all the edges where it is applied at the edge data sources to form DQPe with the tolerance values, max and min (possible values), and the quality measures (metrics) of the dimensions specified by $M=m_{0},\cdots ,m_{n},\cdots ,m_{t}$.

#### 3.3.1. Quality Rules Development

DQPe is, thus, represented by (A,W,D,T,R,M) and is built at the edges using the baseline profile as a blueprint. The dataset is analyzed, and the profile is updated to include data size, attribute count, row count, and attribute details with the maximum, minimum, and percentage of missing values. Quantitative measures of missing data, unique data, and completeness relative to the dataset are added to the quality profile. To guarantee an increase in edge data quality, certain steps must be taken as outlined in the rules. Here is an example of the XML code that describes how to handle missing data in the dataset.

The quality rules of missing data are specified in the XML, as depicted in Figure 3. With quality rule action 2.1, using mean imputation, we replace missing values of attribute *X* with the attribute’s mean, which is calculated using the attribute’s non-missing values. In a normally distributed variable, the median, mean, and mode will all be close to one another. Thus, using the mean or the median to fill in missing data is equivalent, but for skewed attributes having missing value tolerance under 5%, it is recommended to do imputation with the median [53]. Using the mode to fill in gaps in numerical variables is unusual.

When there is a connection between the missing data and the other features of the record, or when there is background knowledge about the likely data values, it is possible to draw conclusions about the missing data and the other features of the record. To fix data issues, supervised machine learning algorithms can be used to predict values and correct missing values. MNAR (missing not at random) means that something is missing in a systematic way rather than by chance. In this case, there is a systematic dissimilarity between the available data and the missing data which can be handled by KNN (K-nearest neighbor). Feedback iterations can aid algorithms in learning and increasing their precision over time. This quality rule is incorporated in rule 2.4. There is uncertainty in the imputed values, so multiple imputations (MI) were proposed in the literature as a method to fill in the gaps. In MI, the required number of imputations is proportional to the frequency of missing data, i.e., more imputations are needed for a dataset that is missing a lot of information [54]. We have incorporated MI in rule 2.6 when the tolerance is above 20% and the data is numerical. As imputation could introduce some bias into the data, it is recommended to remove the row or columns if the missing tolerance is larger, as indicated in rule action id 2.6 and 2.7. Similarly, other DQ rules are specified in XML and sent back to the server, where it is federated.

#### 3.3.2. FDQP Formulation

The FDQP formulation captures and aggregates data quality profiles from multiple sources in federated systems, enabling a comprehensive understanding of data quality at the federated level.
(1)$${\left[DQP\right]}_{Fed}={\left[DQ\right]}_{FedProf}\left(Grp\sum \left({\left[DQP\right]}_{e}\left(A,W,D,T,R,M\right)\right)\right)$$
where $DQP_{Fed}$ represents the data quality profile at the federated level, which is the overall data quality representation obtained through the federated profiling process, ${\left[DQ\right]}_{FedProf}$ (data-quality aware) is the group aggregation of the DQP received from all the edges; the aggregation is based on the dataset and the data quality dimensions specified in the DQP, Grp∑ represents a grouping and aggregation operation applied to the individual data quality profiles ($DQP_{e}$) obtained from the edge data sources, and $DQP_{e}$ represents the data quality profile at the edge level, which is obtained by applying the DQPv (baseline data quality profile) to the edge data sources.

Parameters: *A* (data attributes), *W* (attribute weights), *D* (data quality dimensions), *T* (tolerance levels), *R* (applicable rules), and *M* (quality measures).

Figure 3 gives a comprehensive overview of the FDQP-mandated federation rules. Both global and attribute-based summaries of quality profiles are generated. Attributes and the “MissingData” deletion criteria are both standardized at the federation level and then cascaded to the edges. Additionally, there is another rule that uses federation heuristics to fine-tune the tolerance applied to the attributes.

${\left[DQ\right]}_{FedProf}$ also considers the dimensionality reduction with feature selection where the number of attributes is reduced based on missing data and acceptable tolerance rules. Assume we have a dataset in D-dimensional space with n samples. Dimensionality reduction approaches convert a dataset with dimensionality *D* into a new dataset with dimensionality *d*,while keeping as much of the data’s shape as possible. For each edge, and each attribute *aj*, a missing value *MVi* information vector is created, where *ATol* is the acceptable tolerance for the attribute.
(2)$${\left[MV\right]}_{i}=a_{j}.isNull\cdot count()/D$$
(3)$${\left[{\left[MV\right]}_{i}a\right]}_{i,j}^{n}=\left\{\begin{array}{c} 1,if{\left[MV\right]}_{i}a_{i,j}<\left[ATol_{\left(a_{i,j}\right)}\right]\\ 0,\mathrm{Otherwise} \end{array}\right.$$

Furthermore, the vector is aggregated during ${\left[DQ\right]}_{FedProf}$, considering all the edges where rules are applied based on attribute weight and tolerance. For instance, If the attribute weight is above 0.5, the aggregation rule adds a chosen imputation method; however, if the calculated MVi is 0 (meaning it is above the missing value tolerance) and the attribute weight is insignificant, rules are prescribed to remove the attribute. As a result, these rules override the profile and reduce the dimension to d at the FedProf aggregation. This federated DQP, ${\left[DQP\right]}_{Fed}$, is distributed to the edges, where the rules are applied. All of the data measurements and rules acquired in the preceding profiling’s are considered in the FedProf aggregation. ${\left[DQP\right]}_{Fed}$ will differ from the DQPe in many aspects, including values, size, dimensions, and data metric scores. Federated ${\left[DQP\right]}_{Fed}$ will be used to determine the right quality metric functions to assess a data quality dimension dk for an attribute ai with a weight wj. Edges that fail to meet critical data quality metrics will be dropped from further processing. Finally, the ${\left[DQP\right]}_{Fed}$ is sent to the edge nodes and applied for PSN calculation and evaluation based on different PSN fusion algorithms.

### 3.4. Model Profiling Algorithm

We have detailed the FDQ profiling algorithm in Algorithm 1. This algorithm takes the server and the client processes into account. The number of edges, list of dimensions, list of rules, and version number of the quality profile are the input parameters. The FDQP and quality-enhanced data are the outcomes. Initially, a baseline DQ profile with the quality dimensions and rules list is created and distributed to all edges. Each client creates their local DQP, which is then sent to the profile federation server, where the FDQP is built in accordance with the federation rules. FDQP is further forwarded to clients, where it is applied at the edges depending on DQ tolerances that have been stated in advance. The client is disconnected if the data quality at the edge is not adequate. When applied, the FDQP results in data that is both enriched in quality and useful for further eHealth analytics. For the duration of the data streaming process, iterations are performed, and profile versions change incrementally. Once the profile quality reaches a threshold, the profiling stops and the profile is used in all subsequent iterations. To minimize overhead, only the profile updates are transmitted following the initial run. For continuous real-time profiling, once the edge data are applied with FDQP, the data are marked to be forgotten in the profile, so they will not be used again, and the process is then repeated.

### 3.5. Federated Feature Selection

The FDQP relies heavily on federated feature selection to boost classifier precision. The task of feature ranking [55] entails determining the relative importance of a set of features and then ordering them accordingly. This relative importance is typically determined by a feature selection criterion. Given that the data in our scenario are dispersed across multiple nodes, the challenge is to minimize information loss during federation. The number of available nodes with high-quality data or the varying number of features per node could be the limiting factor, depending on the situation we are tackling. The pseudocode for federated feature selection is provided in Algorithm 2; it ranks features according to criteria including feature value, outlier percentage, and missing data percentage, and these metrics are collectively indicators of data quality.

Given a network with 10 nodes and 100 features per node, we can estimate the difficulty of the ranking combination task because the rankings are not identical.

The federated feature selection is illustrated in Table 3. Column 1 contains the list of features, while columns 2 through 4 contain the aggregated feature selection rank, feature outlier rank, and feature missing rank. The ranking criteria for each column are specified in the header; for example, the feature with the lowest outlier percentage is ranked first (i.e., Rank 1), and the feature with the highest outlier percentage is ranked N, where N is the number of features. The final column, federated feature rank, is calculated as described in Algorithm 2. In the provided features in Table 3 (A, B, C, D, E), D is the best valuable feature (with the least federated rank), while E is the worst one. Priority is given for feature selection if two features have the same federated value, followed by outlier and missing rank. Thus, missing data and outliers have a negative effect on aggregation for making the right decision regarding data selection and feature extraction, as illustrated by the federated feature selection. The federation of feature selection adds value because, in a real-world scenario, each node has only partial information to rank the features as it does not have the entire dataset, making it impossible to accurately compute the importance given to each feature.
**Algorithm 1** Federated Data Quality Profiling Algorithm**Input:**$S_{n}$,

▹ No of Edges
AList,

▹ List of Attributes
DList,

▹ List of Dimensions
RList,

▹ List of Rules
$Q_{Tol}$,

▹ Acceptable Quality Tolerance
v=0
▹ Version of the DQ Profile(DQP)
**Output:**$DQP_{Fed}$
▹ Federated DQ Profile
$PScore_{Fed}$
▹ Federated Patient Similarity Score



                 **//Baseline Profiling**

 1: $DQP_{v}\leftarrow initializeProfile($DList,AList,RList)

▹ Data Quality Profile - Generate
  baseline profile with quality dimensions and rules.



                 **//Baseline Profiling**

 2: ${WLP}_{e}\leftarrow EdgeWorkloadProfileCreate($Config_e,RealTime_e)

▹ Create Workload
  Profile based on config file and edge resource real time parameters.



 3: $DQP_{e}\leftarrow ClientProfileCreate($DQP_v$)+WLP_{e}$

▹ each Edge i



                ** //Federated Profiling**

 4: $DQP_{Fed}\leftarrow DQ_{FedProf}\left(\sum_{e=1}^{n}DQP_{e}\right)$

▹ Federated DQ Profile Aggregation.



                 **//Edge Processing**

 5: $DQEnrichedData_{e}\leftarrow ClientProfileUpdate\left(DQP_{Fed}\right)$

▹ each Edge i

 6: $PScore_{edge}\leftarrow PSNFusionModelpssing\left(DQEnrichedData\right)$

▹ Quality Enriched Data is
  passed to the Edge PSN Model to determine the patient similarity score.



                 **//Federated PSN and Centralised Processing**

 7: $PScore_{Fed}\leftarrow FederatedPSNModelpssing\left(\sum_{e=1}^{n}PScore_{edge}\right)$

 8: **while** ($Accuracy\left(PScore_{Fed}\right)<Q_{Tol}$)

▹ Quality Score Evaluation

 9: v←v+1

▹ Increments Profile Version

10: $DQP_{v}\leftarrow$ DQP_Fed

▹ Baseline Profile is updated with the Federated Profile.

11: Repeat

▹ The process is repeated until the target accuracy is obtained.



                ** //Client process: running on the clients**



12: **procedure**  ClientProfileCreate($DQP_{v}$)

13:       $DQP=GenerateEdgeProfile(DQP_{v})$

▹ DQ Profile

14:       **return**(DQP)

15: **end procedure**



16: **procedure** ClientProfileUpdate($DQP_{Fed}$)

17:       $DQ_{tolerance}\leftarrow extractDQLimits\left(DQP_{Fed}\right)$

18:       **for** $DList_{i}\leftarrow 1,DList_{n}$ **do**

▹ each quality dimension

19:             **if** $DList_{i}\leq DQ_{tolerance}\left[DList_{i}\right]$ **then** disconnectClient()

20:                   **return** 0

21:             **end if**

22:       **end for**

23:       $LFDQP\leftarrow UpdateLocalClientProfile(DQP_{Fed})$

▹ Local Federated DQ Profile

24:       $ClientData_{DQEnriched}\leftarrow ApplyClientDataProfiling\left(LFDQP\right)$

25:       **return** $ClientData_{DQEnriched}$

26: **end procedure**


**Algorithm 2** Federated Feature Selection Algorithm
**Input:**
$S_{n}$,

▹ Participating Edge Source Nodes
$DQP_{Features}$,

▹ Node Features extracted from DQP


$Feature_{Tol}$


▹ Tolerance of number of selected features



**Output:**


$Features_{Fed}$


▹ Federated Selected Features



**//Aggregating Features based on FeatureValue, Outlier and Missing Data from n nodes**



1:**procedure** FederatedFeatureSelection($S_{n}$,$DQP_{Features}$,$Feature_{Tol}$)2:      $Features_{Agg}\leftarrow \sum_{e=1}^{n}DQP_{Features}\left(FeatureValue,Outlier,Missing\right)$3:      $FeatureValue_{Rank}\leftarrow$ sort DESC Features_Agg(FeatureValue)4:      $OutlierData_{Rank}\leftarrow$ sort Features_Agg(Outlier)5:      $MissingData_{Rank}\leftarrow$ sort Features_Agg(Missing)6:      ${\left[Feature\right]}_{Rank}\leftarrow FeatureValue_{Rank}+OutlierData_{Rank}+MissingData_{Rank}$7:      $Features_{Fed}\leftarrow$ sort ${\left[Feature\right]}_{Rank}$ limit by $Feature_{T}ol$8:      **return**($Features_{Fed}$)9:
**end procedure**



### 3.6. Computational Complexity of FDQP

The overall computational complexity of the FDQP approach includes the computational complexity of the FDQP algorithm and the federated feature selection algorithm.

#### 3.6.1. Federated Data Quality Profiling Algorithm (FDQP)

**Baseline profiling:** The complexity of creating the baseline DQ profile (DQPv) is O(D∗W∗R), where *D* represents the number of attributes, *W* represents the number of data quality dimensions, and *R* represents the number of rules.

**Edge profiling:** Creating the edge workload profile ($WLP_{e}$) at each edge node can be assumed to have a complexity of O(1), as it depends on the specific configuration and real-time parameters.

**Federated profiling:** Aggregating the DQPs from multiple edge nodes involves combining the profiles, which can be completed in linear time, resulting in a complexity of O(E∗D∗W∗R), where *E* is the number of edge nodes.

**Edge Processing:** Updating the local client profile ($DQEnrichedData_{e}$) based on the federated profile ($DQP_{Fed}$) can be assumed to have a complexity of O(1), as it depends on the specific update operation.

#### 3.6.2. Federated Feature Selection Algorithm

**Aggregating features** based on feature values, outliers, and missing data from *E* node edges can be performed in linear time, resulting in a complexity of O(E).

**Sorting the aggregated features** based on ranks can be performed in O(NlogN) time complexity, where *N* is the number of features.

**Selecting the top K features** based on the tolerance ($Feature_{Tol}$) can be done in O(N) time complexity.

Thus, the overall complexity of the federated feature selection algorithm is O(NlogN+N), which is = O(NlogN).


**Overall Complexity = Baseline Profiling + Edge Profiling + Federated Profiling + Edge Processing + Federated Feature Selection**


Therefore, the overall computational complexity can be expressed as:O(D∗W∗R+E∗D∗W∗R+NlogN)=O(E∗D∗W∗R+NlogN)

It is linear with the number of edges, and has log-linear complexity with the number of features. It is important to note that the above analysis focuses on the computational complexity of the algorithm itself. Other factors, such as data transfer, network latency, and edge node resources, can also impact the overall performance in edge computing environments. Thorough investigation and benchmarking considering the specific constraints and requirements of the edge environment are necessary to assess the computational demands of the FDQP approach accurately.

### 3.7. DQ XML Profile Illustration

All of our DQPs are written in XML format, which are lightweight and easily readable by humans and machines alike. This makes them incredibly versatile, allowing for the data to be used in various application contexts and shared between any OS platforms at different nodes. A representative XML profile at one of the selected nodes after LDQP processing is shown in Figure 4. The profiling process takes care of the following, which can provide valuable insight into the structure of the data and make it easier to manage DQ.

1.Attribute-wise feature updating that includes maximum value, minimum value, mode, uniqueness, skew, outlier detected, uniqueness, etc. The XML profile structure includes these elements, allowing for a more thorough comprehension of the data.2.Missing value identification and reporting in XML (attribute-wise, overall). It also allows for quick debugging, as well as making it easier to identify and deal with data anomalies.3.How missing values were removed is detailed, using either a specified threshold or specific criteria.4.Missing data imputation rules and the DQ measurements5.Rank the attributes for feature selection according to their relative importance, and the DQ metrics are evaluated.

Figure 5 depicts the federated XML attributes snippets as a subset of the full FDQP attributes in the FDQP model. We can see that according to the FDQP formulation in Section 3.3, only the resulting decisions are highlighted in FDQP. The global threshold value for all the nodes has been established for the delete rows as seen in XML. The selected features are listed according to the algorithm presented in federated feature selection (Algorithm 2), and the data imputation rules for each attribute common to all nodes are presented. After applying FDQP to the data along all of its edges and enforcing constraints in accordance with FDQP’s rules, we remeasure the data metrics to assess how well the profiling worked. We can guarantee that as the data model evolves, so will the associated attribute-level constraints, allowing for a broader range of query and data manipulation capabilities.

## 4. Experimental Evaluation

This section describes the experiments conducted to evaluate our proposed FDQ profiling model. The primary goals are to evaluate the data quality and the accuracy of the training model both before and after the FDQ profiling model is applied to the data at the edges. In the subsections that follow, we will describe the various aspects of the experiments conducted, such as the experimental setup, the dataset employed, the experimental design, and the various scenarios tested. We will conclude by discussing the results, which depict an improvement in accuracy, and the reasoning behind it. Initial assumption: We assume that the data collected at the nodes are homogenous and identically distributed (i.i.d).

### 4.1. Dataset

The dataset we chose for our experiments contains 2126 instances and 23 attributes derived from cardiotocograms, which are continuous measurements of the fetal heart rate using an ultrasound transducer placed on the mother’s abdomen and categorized by expert obstetricians. The parameters used for data analysis are instantaneous fetal heart rate (FHR) and simultaneously communicated uterine contraction signals. The classification results were based on the fetal state labels (N = normal; S = suspect; P = pathologic) [56].

For the selection stage and reduction of attributes, in particular the numerical input data and a categorical (class) target variable, there are two well-known feature selection techniques, mainly ANOVA-f statistics and mutual information statistics. The results of this test can be used for feature selection by removing from the dataset those features that are independent of the target variable.

### 4.2. Experiment Setup

All of our experiments for this study were conducted in IPython, which is enhanced interactive Python software V.3.11.1 for the experimental configuration that includes multiple machine learning libraries for deep learning and federated learning. The entire Fetal Health dataset is randomly divided into five datasets, each corresponding to one of the five edge nodes. The dataset has a few quality issues, so we synthesized errors and noise at the dataset’s edges to reflect a true clinical distribution and to illustrate how FDQP will improve it.

### 4.3. Scenarios—Proposed FDQP Evaluation

To evaluate our proposed FDQ profiling model, we implemented eight scenarios. The first scenario evaluates the data quality profiling with the main focus on accuracy considering baseline accuracy, after missing data imputation, applying local DQP (LDQP), which includes feature selection and rules application. The second scenario illustrates the node selection criteria defined in FDQP, which takes into consideration the completeness and consistency of DQ parameters. In the third scenario, the federated feature selection and ranking are evaluated. The fourth scenario compares the accuracy of the FDQP to that of the LDQP and the baseline. In the fifth scenario, the accuracy, completeness, and consistency of the data quality metrics are examined before and after FDQP. The sixth scenario analyzes the accuracy of various classifiers using FQQP to determine the most accurate classifier. The seventh scenario indicates the number of features selected and the associated training time at each node. Finally, the eighth scenario depicts the PSN accuracy before and after FDQP.

#### 4.3.1. Scenario 1

Create a data quality profile (DQP) based on the XML file containing the dataset and send it to the edge nodes. The quality characteristics of the profile are updated by generating LDQP and sent to the server. As shown in Figure 6, LDQP is applied at the edge node and the experiment’s accuracy is evaluated node-by-node with baseline and after each of the processes (missing data imputation and LDQP).

Node 2 showed the highest increase in accuracy, with a boost from 70.71% to 91.92% after LDQP, and Node 4 exhibited a notable improvement in accuracy, increasing from 65.13% to 89% after LDQP. Node 3 demonstrated a considerable increase in accuracy after rows were removed, with the accuracy rising from 53.66% to 83.53%. This significant enhancement underscores the impact of data preprocessing steps, such as removing rows with missing or incomplete data, in improving the overall accuracy within the FDQP framework.

#### 4.3.2. Scenario 2

Edge node selection occurs before FDQP is applied at the edge node. In other words, LDQP will be used to select nodes on the server. If the DQ metrics values (accuracy, completeness or consistency) are less than the tolerance, the node is eliminated.

One of the first and most important steps in our proposed FDQ profiling is assessing the edge-level distribution of classes and establishing criteria for node selection. Figure 7 illustrates the problem with the consistency and completeness of the Node-3 dataset. The FDQ profiling criteria determine what levels of information must be present in a node’s representation. Accordingly, Node-3 is eliminated from further processing after failing to comply with the DQ requirements.

#### 4.3.3. Scenario 3

The server federates the profile data from the edges. Features are eliminated (feature selection) according to the feature selection algorithm, and the retained features are documented in the feature driven FDQP. The resulting FDQP includes data imputation rules and is transmitted to the edge nodes. In order to assess the PSN’s accuracy and precision, it is necessary to apply FDQP to the edge node. Figure 8 depicts the federated feature rank for each of the attributes in our experimental dataset.

#### 4.3.4. Scenario 4

Evaluation of FDQ profiles is one of the most crucial aspects of our experimentation. Node 3 has already been deleted, and the accuracy after FDQP profiling for the remaining nodes is compared to accuracy after LDQP and baseline accuracy, as seen in Figure 9. We can see that accuracy has improved significantly around 10% with LDQP and to a maximum of 5% with FDQP.

#### 4.3.5. Scenario 5

Before and after applying FDQP, we analyzed the metrics for data quality. Figure 10 shows that the coefficient of variation was reduced following FDQP, suggesting enhanced consistency and that the completeness factor was increased to 100%, indicating that certain data quality issues had been addressed by the process.

#### 4.3.6. Scenario 6

In this scenario, an evaluation is performed using various ML models to determine which classifier is the most accurate in forecasting fetal health at each node, and the results are displayed in Figure 11. Both the random forest classifier and the decision tree classifier have been shown to have the best performance in all of the edge nodes. Edge 2 had the greatest accuracy gain, with random forest classifier attaining 95

#### 4.3.7. Scenario 7

The selection of features is one of the primary characteristics of our suggested FDQP, and its evaluation can be found in Figure 12. We can see that the number of features has been reduced in each of the nodes, which has led to a shorter amount of training time when compared to the initial amount of training time.

#### 4.3.8. Scenario 8

Patient similarity evaluation is performed after FDQP is reviewed, and we observed in Figure 13 that FDQ profiling has unquestionably increased the accuracy, with an average increase of 7% and a maximum gain of 9% accuracy.

## 5. Discussion

The experimental evaluation of our proposed FDQ profiling model provided valuable insights into its effectiveness and impact. The key findings from the experiments are as follows:Improved accuracy: FDQ profiling resulted in an average accuracy increase of 10% with LDQP and a maximum improvement of 15%, attributed to data quality enhancement through missing data imputation, feature selection, and rule-based processing.Enhanced data quality metrics: FDQ profiling significantly improved consistency and completeness, reducing the coefficient of variation and achieving 100% completeness.Effective node selection: FDQ profiling’s node selection criteria successfully identified high-quality nodes, ensuring accuracy improvement by considering completeness and consistency.Classifier evaluation: Random Forest Classifier and Decision Tree Classifier consistently achieved the highest accuracy in fetal health forecasting, suggesting the potential for selecting the best classifier based on dataset and node characteristics.Patient similarity accuracy: FDQ profiling improved patient similarity accuracy by 7% on average and up to 9%, which can have significant implications for various applications, such as personalized medicine and recommendation systems.Profile aggregation and optimized feature selection: Profile aggregation with federated feature selection of attributes from various nodes can improve the efficiency of discriminative features and restrain interference from relatively ineffective features. They are calculated by feature aggregation and then optimized via the proposed rules for elimination, combining the idea of survival of the fittest. The feature with low weight is eliminated in the experiments, leading to improved accuracy and reduced training time.Ensemble-like results: We can see that the results obtained by profiling the data at multiple nodes are, in some cases, more stable with the overall ranking concerning management and resource utilization. The concept of distributing features across nodes and then federating the profile’s results into a final one is analogous to that of ensemble learning or a mixture of experts, and produces more reliable results than a single expert.

In summary, the experimental evaluation showcased the effectiveness of FDQ profiling in enhancing data quality and improving the accuracy of machine learning models. The results not only validate the approach, but also highlight its potential for broader applications across various domains beyond healthcare. By prioritizing data quality components and selecting feature nodes based on relevant metrics, FDQ profiling emerges as a valuable tool for optimizing data quality and advancing machine learning model classification.

The FDQP approach proposed in this paper is founded on several underlying principles that make it a promising technique for evaluating the quality of patient data collected from edge nodes in federated environments. Understanding these principles is essential to value the relevance and potential of FDQP across various domains.

Federated learning and data quality profile (DQP): The foundation of FDQP lies in the federated learning paradigm, which enables collaborative model training across distributed edge nodes without centralizing raw data. By preserving data privacy at the edge, FDQP addresses the challenges of data silos and privacy concerns in healthcare and other domains. The DQP encapsulates data quality dimensions, defining a framework to assess the quality of data attributes, such as completeness, accuracy, and consistency. FDQP encapsulates DQPs from different edge nodes into a unified formal model. This formalization enables seamless comparisons, aggregation, and analysis of data quality metrics, facilitating better decision making during model aggregation in federated learning.

Federated feature selection: FDQP employs federated feature selection, which combines local feature selections at each edge node and global feature ranking. This technique enhances the precision of classifiers by selecting relevant features while mitigating the impact of noisy or irrelevant attributes. The use of outlier percentage and missing data percentage as criteria in feature selection makes FDQP robust to variations in data quality across different edge nodes.

Lightweight profile exchange: FDQP introduces a lightweight profile exchange mechanism based on XML that shares summary statistics of data quality attributes among edge nodes. This exchange avoids the transmission of raw data, optimizing data quality achievement, and improving the overall efficiency of the federated learning process. This approach is particularly relevant in resource-constrained edge computing environments.

Enhancing data quality and accuracy: The primary motivation behind FDQP is to enhance data quality and, consequently, the accuracy of federated learning models. By identifying and quantifying data quality issues at the edge nodes, FDQP enables targeted data cleaning, outlier detection, and imputation strategies. This results in improved data quality, reducing the impact of noisy or biased data on the global model’s performance.

Scalability and efficiency: FDQP addresses the scalability and efficiency challenges of traditional centralized data quality assessment methods. By conducting quality evaluations locally at the edge nodes and sharing aggregated quality metrics, FDQP preserves privacy and reduces the risk of data breaches.

### Limitations

While our current work focuses on addressing the challenges of federated data quality profiling in edge computing environments, there are still a few challenges that warrant further investigation and future research.

Generalizability: The methodology’s validation on a specific fetal dataset raises concerns about its generalizability to other types of patient data and different domains. Further research, including different types of data and real-world scenarios, would provide a more comprehensive evaluation of FDQP’s effectiveness.

Trustworthiness of edge nodes: The assumption of reliable and trustworthy edge nodes might not always align with real-world scenarios. FDQP should be examined under situations where edge nodes might be unreliable or malicious to ensure the approach’s robustness and security.

Limited quality dimensions: The current implementation of FDQP focuses on specific data quality dimensions such as accuracy, completeness, and consistency. Extending the model to incorporate additional quality dimensions could provide a more comprehensive evaluation.

Privacy and security: While FDQP incorporates security measures through federated learning, it might not address extremely stringent privacy and security requirements, particularly in high-sensitivity data scenarios. Future research should explore additional privacy-enhancing techniques to accommodate diverse security needs.

By recognizing these remaining challenges and conducting further research to address them, the FDQP approach can be refined to enhance its applicability, robustness, and credibility. This will facilitate its adoption in various domains, leading to improved data quality, more accurate machine-learning models, and better decision-making processes.

## 6. Conclusions and Future Work

To ensure reliable and meaningful insights from any data analytics process, quality is a determining factor of utmost importance. The end-to-end process of data integrity is required to achieve DQ, which is attained through a meticulous process of data cleansing and data governance. Poor data, on the other hand, will have an adverse effect on ML classification results because it will lead to poor analytics. In conclusion, we have demonstrated that FDQ profiling can handle various quality issues at multiple edges while keeping data localized, thereby enabling data privacy and reducing data resource consumption and transportation costs. Erroneous data reduce the performance of learning algorithms, but this is the first time federated profiling has been used to mitigate the consequences of improving data quality and enhancing ML classification at multiple edges. In our future research, we plan to extend the application of FDQ profiling beyond healthcare and explore its synergies with other domains, such as finance, e-commerce, manufacturing, and more, with the goal of optimizing data quality and decision-making processes in diverse industries. By embracing this interdisciplinary approach, we aim to contribute to the advancement of data quality management practices and drive innovation in various fields. In addition, we will explore potential advancements on edge node data compression and PSN similarity as a powerful tool for data imputation, which has the potential to significantly enhance data quality. 

## Figures and Tables

**Figure 1 sensors-23-06443-f001:**
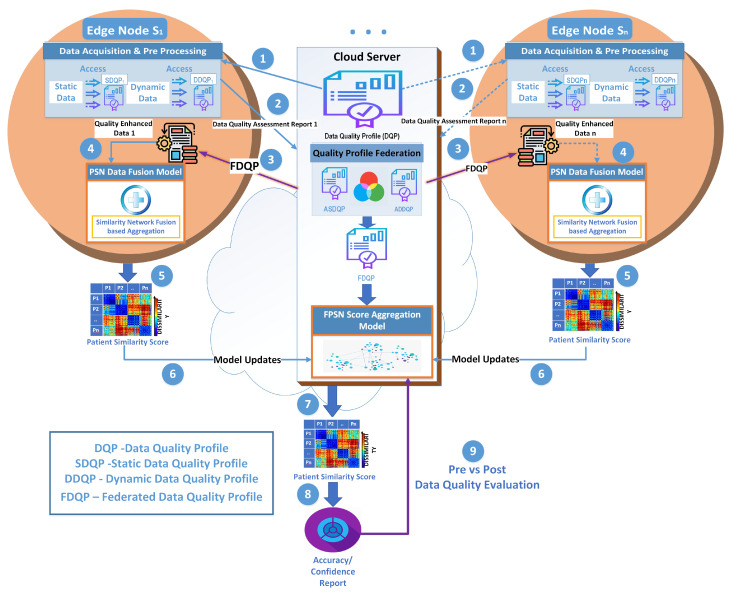
FPSN enhanced by FDQP at the edge.

**Figure 2 sensors-23-06443-f002:**
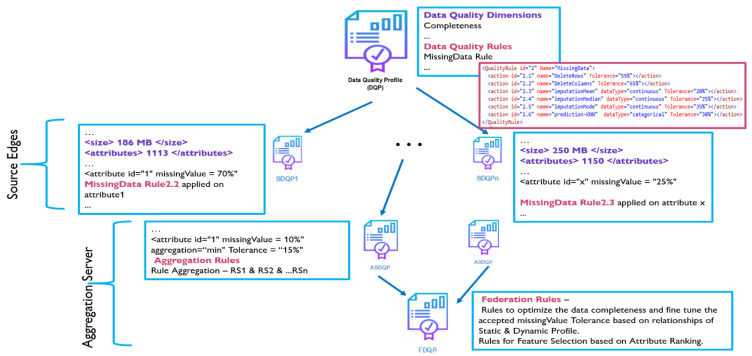
FDQ profiling example.

**Figure 3 sensors-23-06443-f003:**
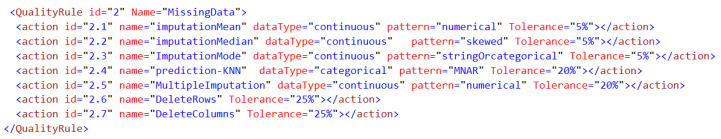
XML quality rule for missing data in baseline DQP.

**Figure 4 sensors-23-06443-f004:**
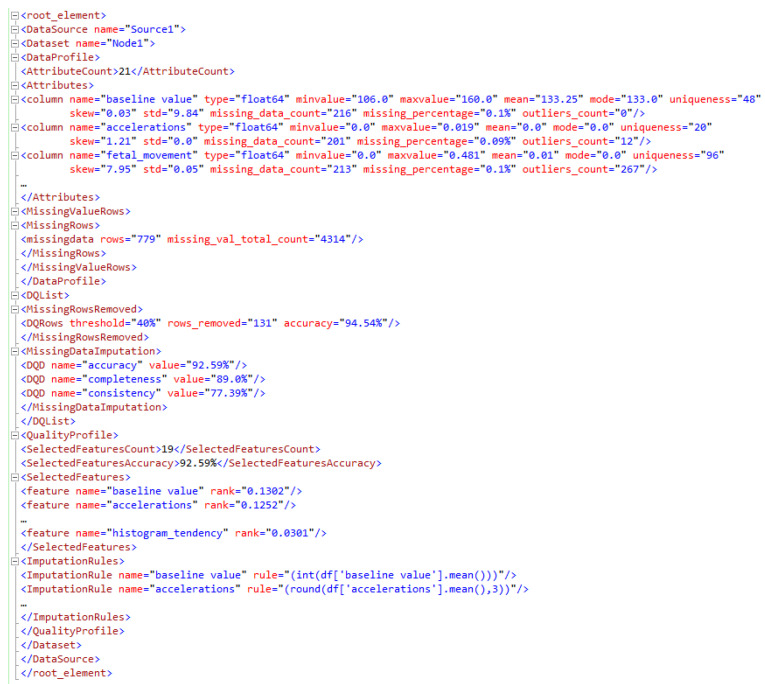
Node 1 consolidated LDQP-snippet.

**Figure 5 sensors-23-06443-f005:**
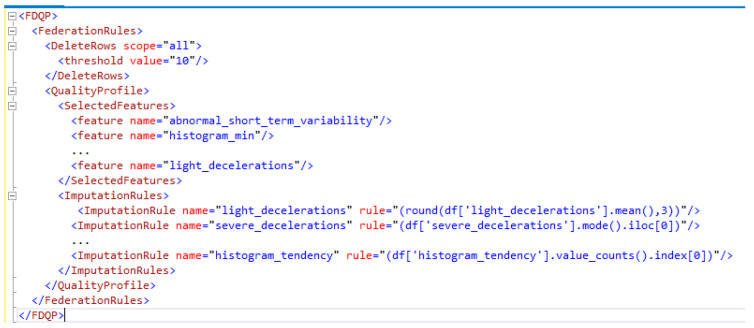
FDQP XML snippet.

**Figure 6 sensors-23-06443-f006:**
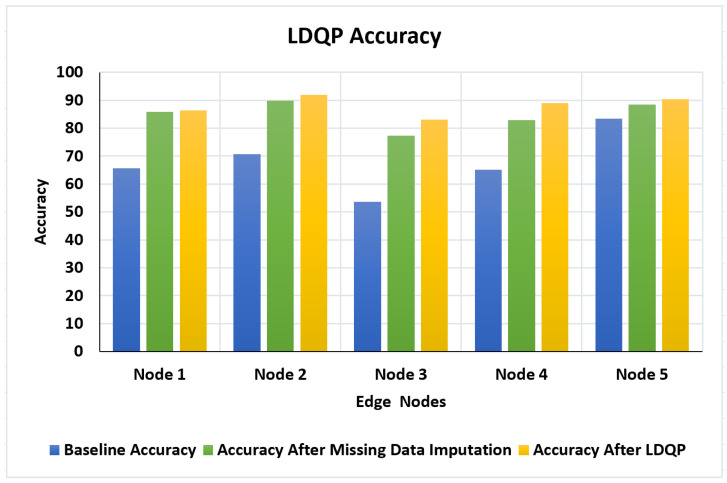
Local data quality profile (LDQP) accuracy evaluation.

**Figure 7 sensors-23-06443-f007:**
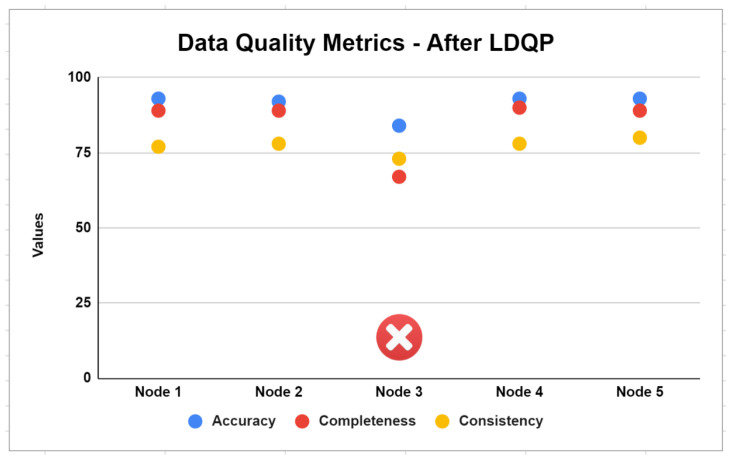
Node selection based on LDQP data quality metrics.

**Figure 8 sensors-23-06443-f008:**
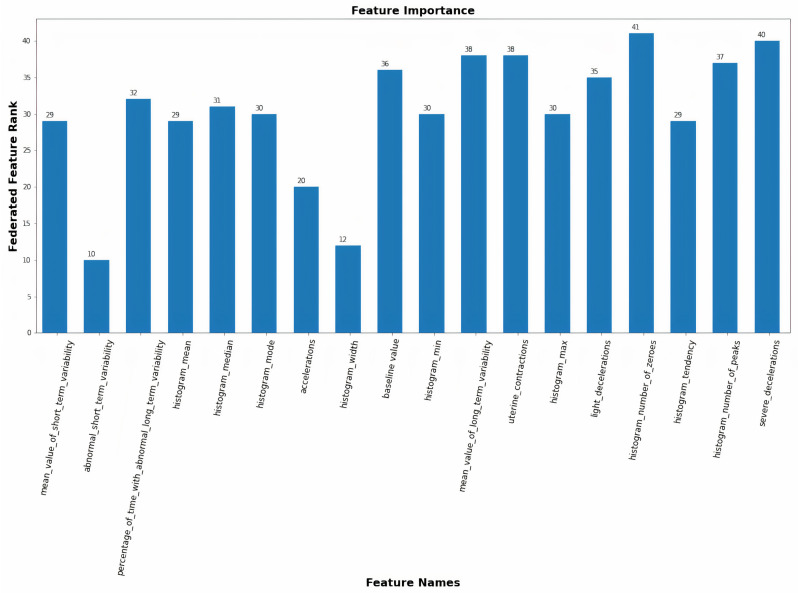
Federated feature selection.

**Figure 9 sensors-23-06443-f009:**
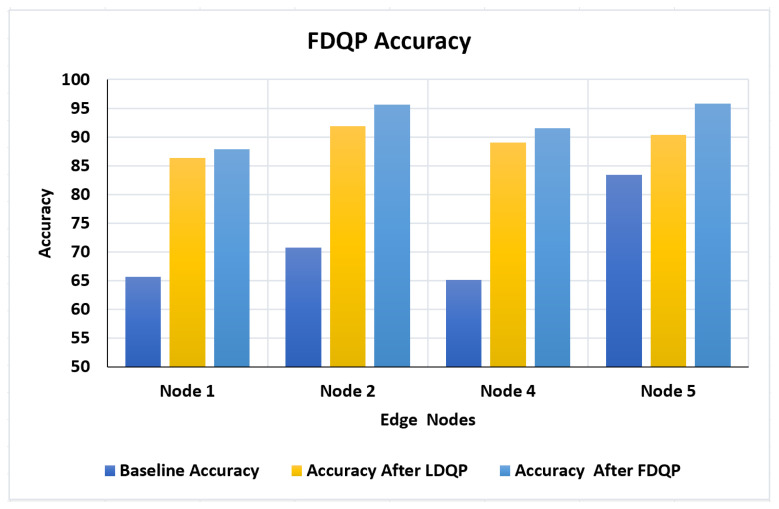
Accuracy (baseline, after LDQP, and after FDQP).

**Figure 10 sensors-23-06443-f010:**
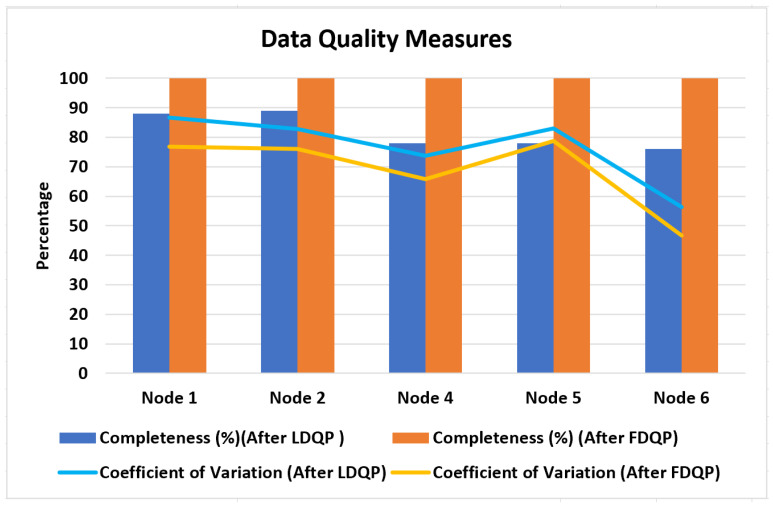
Data quality metrics assessment after LDQP and FDQP.

**Figure 11 sensors-23-06443-f011:**
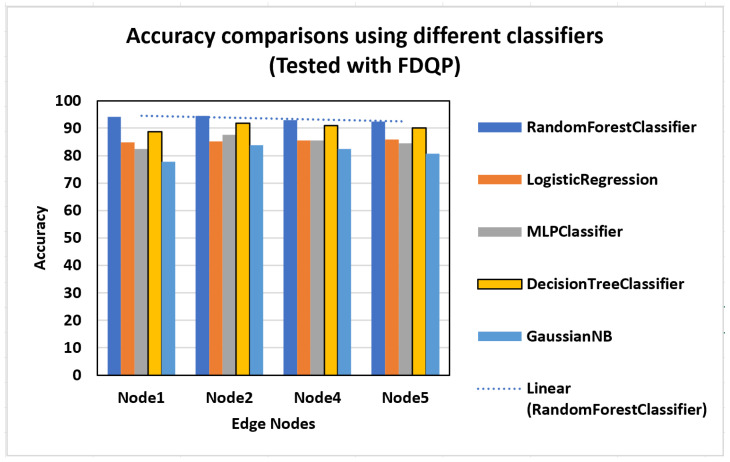
Accuracy comparisons with different classifiers.

**Figure 12 sensors-23-06443-f012:**
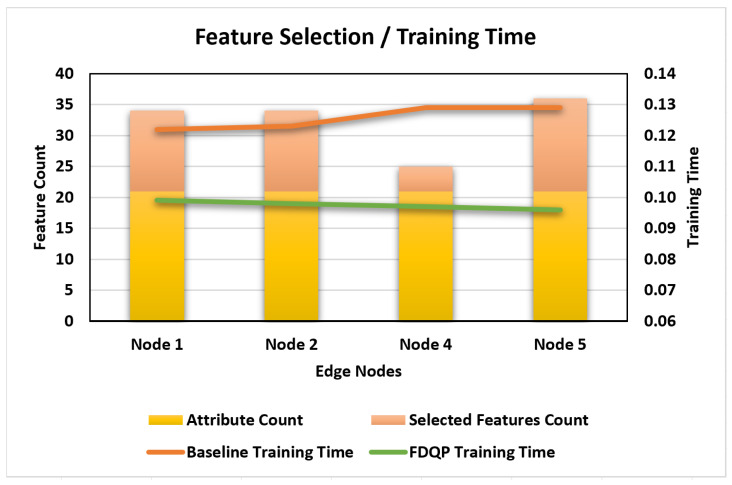
FDQP feature selection vs. training time.

**Figure 13 sensors-23-06443-f013:**
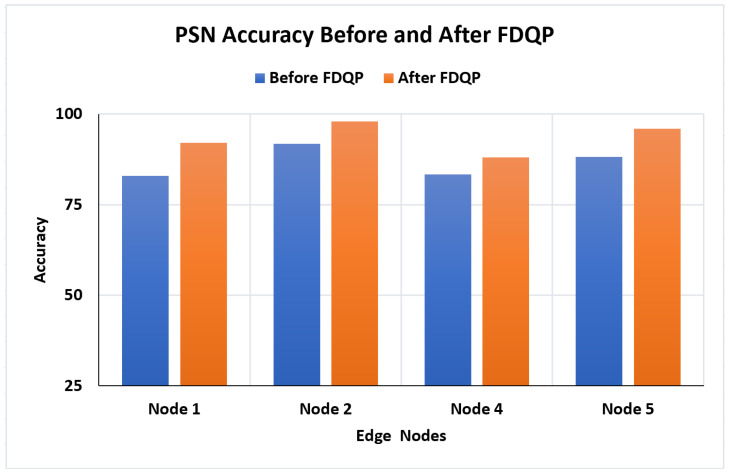
PSN accuracy before and after federated data quality profiling.

**Table 1 sensors-23-06443-t001:** Summary of related work.

Concept	Reference	Key Features	Advantages (+)/Disadvantages (−)
**FL for Data** **Protection**	[17]	Uses FL for protecting clinical information, eliminates data transfer, maintains data confidentiality.	+ Preserves data privacy, allows local storage of patient data. − Challenging to train clients with poor or biased data.
**DQ** **Assessment**	[7]	Discusses DQ profiling, eliminates false positives, improves DQ evaluation.	+ Enhances DQ evaluation, maintains data confidentiality. − Does not address federated or distributed data specifically.
**Federated Sensor** **Service Clouds**	[19]	Proposes DQ-centric architecture for federated sensor service clouds, exchanges high-quality data in real time.	+ Enables real-time data exchange, improves data quality. − Challenges with training clients having poor or biased data.
**Quality-aware** **User** **Recruitment** **in MCS**	[30]	Introduces quality-aware user recruitment in MCS using FL, predicts the quality of sensed data.	+ Optimizes observed data quality, improves FL performance. − Limited to the MCS context.
**FL for In-Edge** **AI**	[21]	Presents an in-edge AI framework using FL, boosts computation and communication per edge device, handles energy consumption, privacy, and fairness challenges.	+ Improves computation and communication efficiency, addresses energy and privacy concerns. − Focuses on edge computing, may have limitations in scalability.
**Fairness in FL**	[22]	Introduces FairFed, an FL framework that rejects model updates leading to attacks, ensures fairness in FL.	+ Enhances security and fairness in FL, prevents model poisoning attacks. − Does not explicitly address DQA.
**Data** **Heterogeneity** **in FL**	[39]	Proposes FedACS framework to address data heterogeneity in FL, generates insights about local data and forms a diverse client pool.	+ Handles data heterogeneity, improves FL efficacy. − Does not focus on DQA directly.
**Communication** **Strategies in FL**	[46]	Discusses communication strategies in FL, investigates the impacts of device-centric communication architecture.	+ Considers realistic communication scenarios, improves efficiency. − Does not specifically address DQ concerns.
**Federated Edge** **Learning**	[51]	Uses DQ aspects for scheduling edges in federated edge learning, prioritizes reliable devices with diverse datasets.	+ Enhances edge selection in FL, considers DQ for scheduling. − Limited to federated edge learning context.
**Concept Drifts** **in FL**	[52]	Introduces the adaptive-FedAVG FL approach to handle concept drifts in nonstationary data-generating systems.	+ Adapts to concept drifts, improves model flexibility. − Focuses on concept drift, may not cover other data streaming concerns.
**Client Selection** **Policy**	[37]	Proposes an auction, client selection policy embedded in a neural network.	+ Reinforcement learning- based client selection. − May require additional computational resources.
**Accuracy** **Threshold** **Aggregation**	[35]	Privacy-preserving, multi-party data learning.	+ Reduces privacy leakage probability. − Requires threshold tuning.

**Table 2 sensors-23-06443-t002:** Comparison of the proposed model against the state-of-the-art approaches.

Reference	Approach	Node/ Client Selection	Dimension Reduction/ Feature Selection	Data Heterogeneity	Federated Profiling/ Aggregated Models	DQ Aware	FL Based	Accuracy Improvement
**[30]**	Quality-aware user selection	✓	-	-	-	✓	✓	-
**[31]**	FedTriNet	-	-	-	-	-	✓	-
**[32]**	Federated Shapley value	-	✓	-	-	-	✓	-
**[34]**	FDSS	✓	-	-	-	-	✓	-
**[37]**	AUCTION	✓	-	-	-	✓	✓	-
**[50]**	FAIR	-	-	-	✓	✓	✓	
**[38]**	TiFL	✓	-	✓	-	-	✓	-
**[39]**	FedACS	✓	-	✓	-	-	✓	-
**[40]**	Ada-FedSemi	-	-	-	✓	-	✓	✓
**[DQA-FDQP]**	Proposed FDQP	✓	✓	✓	✓	✓	✓	✓

**Table 3 sensors-23-06443-t003:** Illustration of federated feature selection.

Federated Feature Selection				
**Ranking** **Criteria**	**Best Rank: 1** **(Most Feature Value)** **Worst Rank: N**	**Best Rank: 1** **(Least Outlier %)** **Worst Rank: N**	**Best Rank: 1** **(Least Missing %)** **Worst Rank: N**	**Best Rank: 1** **(Most Valuable)** **Worst Rank: N**
**Feature (N)**	**Aggregated Feature** **Selection Rank**	**Aggregated Feature** **Outlier Rank**	**Aggregated Feature** **Missing Rank**	**Federated** **Feature Rank**
A	3	1	8	12
B	5	4	7	16
C	9	8	2	19
D	1	2	5	8
E	7	23	13	43
...	...	...	...	...

## Data Availability

The dataset is available at UCI Machine Learning Repository, https://archive.ics.uci.edu/ml/datasets/Cardiotocography (accessed on 9 June 2023).

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
