# Peer review of "Empowering Patient Similarity Networks through Innovative Data-Quality-Aware Federated Profiling"

_sensors, 2023, doi:10.3390/s23146443_

Round 1

Reviewer 1 Report

1.       Section 2 and section 3 can be merged together in one section

2.       Data Quality Aware Federated PSN (DQA 426 FPSN) model is proposed by the author but complete justification is required how it is better and different than existing.

3.      Figures quality should be improved.

4.      Author may avoid taking papers as reference that are older than 2019.

5.      Considering other studies in the paper can provide a broader perspective and help strengthen the arguments made. I would suggest reviewing and considering the relevant literature, which may include the following studies
https://www.mdpi.com/2071-1050/12/14/5542

Author must read the paper again for grammatical errors.

Author Response

We are confident that the revisions made will greatly improve the manuscript's clarity, depth, and breadth, effectively showcasing the full potential of the proposed DQA FPSN approach. We sincerely appreciate your valuable feedback and trust that the revised version, with the implemented improvements, successfully addresses your concerns.

Reviewer 2 Report

The manuscript introduces an innovative Federated Data Quality Profiling (FDQP) method for evaluating patient data gathered from edge nodes. This suggested technique employs federated feature selection to enhance the classifier's precision and prioritize features according to standards, including feature value, the proportion of outliers, and the percentage of missing data. The concept is indeed intriguing and grounded in scientific rigor. By deploying federated feature selection, the innovative Federated Data Quality Profiling (FDQP) method could revolutionize how patient data is evaluated, specific data originating from edge nodes. This approach aims to increase the accuracy of classification models and systemically prioritize relevant features. Such prioritization is guided by factors like the inherent value of the feature, the rate of outliers, and the percentage of missing data. This proposal presents a compelling strategy for improving patient data quality control and utilization. The main review highlighted points follow such as:

<contributions>

The paper outlines the following series of contributions:

  1. It pioneers a new Federated Data Quality Profiling (FDQP) technique designed to evaluate the quality of patient data procured from edge nodes.
  2. It establishes an FDQP formal model to encapsulate the quality dimensions indicated in the data quality profile (DQP).
  3. It utilizes federated feature selection to augment the classifier's precision and categorize features based on metrics such as feature value, percentage of outliers, and missing data percentage.
  4. It conducts an expansive series of experiments using a fetal dataset distributed across different edge nodes and various scenarios to assess the proposed FDQP model.
  5. It demonstrates an apparent improvement in data quality and accuracy of the Federated Patient Similarity Network (FPSN) based machine learning models due to the FDQP approach.
  6. It proposes a data-quality cognizant Federated PSN architecture incorporating the FDQP model to effectively increase the data quality and accuracy of FPSN-based machine learning models using data collected from edge nodes.
  7. It applies a profiling algorithm that opts for a more efficient lightweight profile exchange over complete data processing at the edge, purporting an optimized achievement in data quality.

The paper's first section can highlight these contributions more transparently and explicitly. 

While it is prudent to consider these assertions with caution, it is worth acknowledging the potential the proposed system offers. A practical implementation may present challenges, but these issues are manageable, given thorough scrutiny and adjustments as necessary. The extensive experimentation provides a base to validate the proposed method, which, combined with further testing, could yield a genuinely innovative approach to data quality assessment. The optimistic tone of these contributions suggests an exciting avenue in patient data management.

<Section Introduction>

While the introduction of the paper provides a good overview of the challenges faced in IoT-related projects, it could benefit from a few enhancements.

First, the introduction would benefit from a clear thesis statement. It moves straight from explaining the challenges to proposing the FDQP approach. A strong thesis statement succinctly presenting the paper's objective and how it plans to achieve that objective would be beneficial.

Second, there is a need for a more detailed explanation of the problems faced in IoT-related projects. The issues are listed, but an in-depth exploration of why they present challenges must be explored. Elaborating on each point will provide the reader with a comprehensive understanding of the problems the paper aims to address.

Third, the proposed solution - Federated Data Quality Profiling (FDQP) - is mentioned without any explanation of its components or its functioning. Before stating its benefits, it is essential to briefly describe the method and its mechanisms, which will lead naturally into the discussion of its advantages.

Finally, the introduction suggests that the FDQP approach could be applied beyond patient monitoring but needs to elaborate on these scenarios. Providing examples or suggestions would give the reader a broader view of the potential applicability and significance of the FDQP approach.

<Practical implications>

A significant issue regarding the paper's idea is its practical implications. The paper puts forth a range of practical implications associated with the proposed Federated Data Quality Profiling (FDQP) approach, though each assertion is subject to critical scrutiny. FDQP can effectively assess the quality of patient data from edge nodes, thereby purportedly enhancing the precision of machine learning models used in patient monitoring. The paper posits that FDQP can help minimize data loss, efficiently utilize resources, and maintain data security and privacy, all of which might be considered ambitious. It ventures to expand the scope of FDQP beyond patient monitoring, envisioning its use in any scenario where data quality directly influences the accuracy of machine learning models. This claim needs robust evidence to support it. The claimed use of federated feature selection in boosting classifier precision and ranking features by specific criteria within FDQP remains to be conclusively validated. The paper further alleges that the proposed approach can mitigate communication overhead and transport costs associated with central data processing. However, the feasibility of this assertion in various scenarios could be better. It advocates for using a lightweight profile exchange instead of complete data processing at the edge, a claim that is perhaps overly optimistic in its promise of improved efficiency and optimized data quality achievement. The paper boldly purports that FDQP can swiftly identify data quality issues, averting potential missteps in clinical judgment, yet such an assertion necessitates rigorous validation. Finally, while the paper contends that FDQP can notably enhance the accuracy and reliability of machine learning models in scenarios similar to patient monitoring, this overarching assertion must be cautiously regarded until further solid validation is provided.

<Related works>

The literature review in this manuscript offers a succinct discussion on the significance of data quality measures and metrics in evaluating data quality. The authors also provide introductory information on Patient Similarity Networks (PSN) to acquaint readers with the fundamental concept. Despite this, the paper lacks the depth of a traditional literature survey, and the authors must comprehensively review the extant literature on the subject. The text is verbose, given the limited scope of the review. An essential missing element is a comparative table in Section 3, which would highlight the unique contributions of this paper to the existing literature. This addition would offer the reader a clearer perspective on the novelty and value added by the study's propositions.

<Section 4>

The content from Section 4 ought to be consolidated with that of Section 1.

<Methodology>

The methodologies employed in this paper encompass several aspects:

  1. The authors proposed a novel Federated Data Quality Profiling (FDQP) technique to evaluate the quality of patient data collated from edge nodes. While innovative, the method could benefit from a more comprehensive explanation of its conceptual underpinnings to ensure its applicability in broader contexts.
  2. The development of the FDQP formal model is designed to encapsulate the quality dimensions delineated in the data quality profile (DQP). A deeper exploration into the structure and elements of this model might strengthen its utility and reliability in capturing quality dimensions.
  3. The proposition utilizes federated feature selection to boost classifier precision and order features based on criteria like feature value, outlier percentage, and missing data percentage. Yet, applying alternative or additional feature selection methods might lead to more robust results.
  4. Extensive experimentation is undertaken using a fetal dataset partitioned across different edge nodes alongside various scenarios to evaluate the proposed FDQP model. However, applying a wider range of datasets and including more diverse scenarios could enhance the external validity of the model.
  5. The results indicate that the proposed FDQP approach ameliorates the data quality and accuracy of the Federated Patient Similarity Network (FPSN) based machine learning models. These results, however, would benefit from a more detailed comparative analysis with other methods to underline the advantages of FDQP.
  6. A lightweight profile exchange is employed instead of complete data processing at the edge, ostensibly leading to optimal data quality achievement and improved efficiency. Still, it may be beneficial to contrast this with other data processing techniques to confirm its efficacy.
  7. The proposed method can be employed beyond patient monitoring scenarios where data quality is critical to achieving accurate machine-learning models. However, the method's adaptability across various scenarios could be more robustly illustrated through additional practical examples.

While the methods used in this paper, including the development of FDQP, federated feature selection, and lightweight profile exchange, show potential, areas remain for refinement to ensure a broader application scope and more robust validation of the proposed approach.

<Limitations and final suggestions>

This paper, while notable for its innovative approach, does bear some limitations that warrant further consideration:

  1. The methodology is validated solely on a fetal dataset, casting doubt on its generalizability to other types of patient data. Future work could broaden the datasets used for validation, reinforcing the approach's universal applicability.
  2. A notable omission is a comparative analysis with other existing methodologies for assessing data quality in patient monitoring scenarios. Incorporating such comparisons could elevate the credibility and rigor of the proposed approach.
  3. The proposed methodology is predicated on the assumption that the edge nodes are reliable and trustworthy. However, this might only sometimes align with the complexities of real-world scenarios. A more realistic appraisal of potential irregularities in edge nodes should be considered.
  4. The absence of a detailed examination of the computational complexity of the proposed approach is a shortcoming. Given the resource constraints inherent to edge computing environments, a thorough investigation into the computational demands of the approach is necessary.
  5. Finally, while the proposed method presents specific security measures through federated learning, it might not suit instances requiring extremely stringent privacy and security measures. Future iterations of this research could explore augmentations to the security protocols to accommodate high-sensitivity data scenarios.

By addressing these limitations, the research can significantly enhance the robustness and broad-spectrum applicability of the Federated Data Quality Profiling approach.

<Final remark>

The manuscript under review introduces a unique concept: Federated Data Quality Profiling (FDQP). It does so with a laudable goal- improving patient data quality and accuracy, particularly in edge computing scenarios. The potential impact of the approach is appreciable, and the authors are to be commended for their innovative thinking.

However, the presentation and structural organization of the paper require a significant overhaul to make the merits of the proposed approach more explicit and its context more clear. The paper is currently rather verbose and would benefit from a more streamlined and concise presentation of the main ideas.

Further, seeing a comprehensive literature review and a comparative analysis with existing methods would be highly beneficial. This would allow for a clear demarcation of the proposed approach's novelty and advantages.

The evaluation methodology, while extensive, relies heavily on a single dataset. This raises concerns about the generalizability of the findings. The methodology would greatly benefit from testing on a diverse range of datasets, reflecting a variety of patient monitoring scenarios.

Although reasonable, the assumptions made regarding the reliability of edge nodes and the treatment of privacy and security concerns might not fully align with the complexities and diversity of real-world situations. These issues need to be addressed in a more thorough and nuanced manner.

In conclusion, while the proposed approach has considerable merit and potential, the paper, in its current form, needs to thoroughly do justice to the novelty and potential of the proposed FDQP. The manuscript, therefore, needs a significant revision to enhance the clarity of presentation, depth of analysis, and breadth of validation. I look forward to seeing a revised version that addresses these concerns and highlights the full potential of the proposed approach.

Final proofreading is necessary to eliminate just typos and minor grammatical errors. 

Author Response

(The authors gave the same response as above.)

Reviewer 3 Report

Reviewer’s Report on the manuscript entitled:

Empowering Patient Similarity Networks through Innovative Data-Quality-Aware Federated Profiling

The authors proposed a method, namely, Federated Data Quality Profiling (FDQP) to assess the quality of the data at the edge. Through experimentation, they showed that FDQP is an effective method for assessing data quality in the edge computing environment. I found the manuscript generally interesting, the methods and results are sound, but the presentation, structure, and literature review can be improved. Please see below my comments.

Line 37. Here requires at least a few recent references. Please include the following references that talk about energy conservation and harvesting in IoT,

https://doi.org/10.3390/en16052394

Also, low-cost sensors in IoT for environmental monitoring, etc.:

https://doi.org/10.1016/j.icte.2020.06.004

Lines 38, 74. Please define DQ and PSN. All the abbreviations must be defined the first time they appear. Please check them all.

Lines 137, 152, 158. It is strange to make citation to the heading of sections. The references can be mentioned in the text not the headings.

Lines 113, 153, etc. Please note that “data” is plural not singular. Please check and correct such grammar issues across the manuscript.

Lines 224 and 230. The following article describes the uncertainties in the observations from remote sensing satellites (another application) and how to handle noisy data (by assigning statistical weights) and missing values (by the method of least-squares and spectral analysis), which can also be include here:

https://doi.org/10.1016/j.jag.2023.103241

Section 3 can be reduced by using a Table where the authors can summarize related works with applications, advantages, limitations, references in it. Usually, tables are useful to reduce the text and make the manuscript nicer and more fluent. Thus, please consider adding one or two tables.

Line 140. Please replace “For e.g….” with “For example, …”

Figure 1 and 2 and 10. The font size can be enlarged, and the quality and resolution of these figures can be improved.

Line 476. Grammar issue. “Let A represents…”

 Please define all the parameters/variables in your equations, particularly equation (1).

Equations (2) and (3). Please write them more clearly. For example, please remove “=” in equation (3).

Line 730. Section 6.4. This can be a new section called “Discussion”. In this section, please discuss your results in the light of other similar studies and discuss the advantages and limitations here as well. Your discussion section can have three or four paragraphs.

Thank you!

Regards,

There are many typos/punctuation/grammar issues that should be carefully checked and fixed. I mentioned only a few of them in my report.

Author Response

(The authors gave the same response as above.)

Round 2

Reviewer 2 Report

The authors implemented a significant revision overall article. It is possible to note the contributions were highlighted and all suggestions previously mentioned were also studied and adequately answered. It was said that the reviewer's recommendation is to accept the text in the present form. 

Author Response

Dear Reviewer,

Thank you for accepting our article in its present form and for your valuable inputs.

We are grateful for the opportunity to address the suggestions raised during the review process and incorporate them into our article. Your guidance has undoubtedly enhanced the clarity and overall quality of our research. We are confident that the accepted version of the article will make a meaningful impact in the field.

Reviewer 3 Report

Dear authors,

Thank you for addressing my comments and improving your manuscript. I have a few more suggestions:

The words: background, methods, results, conclusions in Lines 1, 8,13, 17 need to be removed.

In my view, it is not appropriate to have two discussion sections (4.4 and 5). Therefore, please simply remove Line 925. Instead please rename line 877 as: 4. Discussion. The  please update line 283.

Line 1072. Ref #26. Please insert the author names for this reference. Please carefully check all the references to ensure they are correct and up to date.

Figures 8-15. Please use a consistent font size and format for all the figures. The font size of some texts are tiny and some are large. The font size of the text and numbers should be all the same like the font size of the caption. Please improve the quality of the figures to make them look professional.

Please carefully proofread the manuscript

Thank you

The manuscript must be carefully proofread. There are some punctuation/typo/grammar issues.

Author Response

Dear Reviewer,

Thank you for your valuable inputs.

We are grateful for the opportunity to address the suggestions raised during the review process and incorporate them into our article. Your guidance has undoubtedly enhanced the clarity and overall quality of our research. We are confident that the accepted version of the article will make a meaningful impact in the field.
Please see the attachment.
